# Application of Experimental Measurements in a Wind Tunnel to the Development of a Model for Aerodynamic Drag on Elite Slalom and Giant Slalom Alpine Skiers

**Matej Majerič [1], Nina Verdel [2], Jan Ogrin [1], Hans-Christer Holmberg [3,4] and Matej Supej [1,2,\***

[1] Faculty of Sport, University of Ljubljana, 1000 Ljubljana, Slovenia; matej.majeric@fsp.uni-lj.si (M.M.); jan.ogrin@fsp.uni-lj.si (J.O.)
[2] Department of Health Sciences, Mid Sweden University, 831 25 Östersund, Sweden; nina.verdel@miun.se
[3] Department of Physiology and Pharmacology, Biomedicum C5, Karolinska Institute, 171 77 Stockholm, Sweden; hans-christer.holmberg@ki.se
[4] Department of Health Sciences, Luleå University of Technology, 971 87 Luleå, Sweden
**\*** Correspondence: matej.supej@fsp.uni-lj.si; Tel.: +386-1-520-77-00

**Abstract:** Aerodynamic drag is a major cause of energy losses during alpine ski racing. Here we developed two models for monitoring the aerodynamic drag on elite alpine skiers in the technical disciplines. While 10 skiers assumed standard positions (high, middle, tuck) with exposure to different wind speeds (40, 60, and 80 km/h) in a wind tunnel, aerodynamic drag was assessed with a force plate, shoulder height with video-based kinematics, and cross-sectional area with interactive image segmentation. The two regression models developed had 3.9–7.7% coefficients of variation and 4.5–16.5% relative limits of agreement. The first was based on the product of the coefficient of aerodynamic drag and cross-sectional area ($C_d \cdot S$) and the second on the coefficient of aerodynamic drag $C_d$ and normalized cross-sectional area of the skier $S_n$, both expressed as a function of normalized shoulder height ($h_n$). In addition, normative values for $C_d$ ($0.75 \pm 0.09$–$1.17 \pm 0.09$), $S_n$ ($0.51 \pm 0.03$–$0.99 \pm 0.05$), $h_n$ ($0.48 \pm 0.03$–$0.79 \pm 0.02$), and $C_d \cdot S$ ($0.23 \pm 0.03$–$0.66 \pm 0.09$ m$^2$) were determined for the three different positions and wind speeds. Since the uncertainty in the determination of energy losses due to aerodynamic drag relative to total energy loss with these models is expected to be <2.5%, they provide a valuable tool for analysis of skiing performance.

**Keywords:** biomechanics; mechanical modelling; energy; dissipation; energy loss; coefficient of aerodynamic drag; cross-sectional area; performance; GNSS; GPS

## 1. Introduction

The execution of turns in connection with the four major events of the Olympic sport of alpine skiing, i.e., slalom, giant slalom, super-G (super-giant slalom), and downhill [1], differs considerably, due to large differences in the placement of the gates, the length of each turn, the incline of the slope, and, above all, skiing speed [2]. In an attempt to maximize speed, skiers try to minimize the dissipation of mechanical energy [3,4], which is caused primarily by aerodynamic drag and ski–snow friction (which increases when the skis are not guided in a manner that avoids skidding) [1,5].

The extent to which different skiers expose themselves to aerodynamic drag varies considerably [6], being the cause of almost 50% of the difference in race time between slower and faster skiers [7]. As early as 1983, Leino and colleagues [5] estimated that aerodynamic drag is responsible for 10–40% of energy losses associated with alpine skiing. This form of resistance exerts its most pronounced impact on performance in the speed disciplines (Super-G and downhill), particularly as the speed increases [8], contributing to more than 80% of total resistive force [9]. At the same time, Meyer, Pelley, and Borrani [10] demonstrated that usage of a more dynamic, rather than a compact technique when

performing giant slalom is associated with 10% greater energy loss due to aerodynamic drag. Although not the primary determinant of the performance of elite giant slalom skiers, aerodynamic drag accounts, on average, for 15% (range 5–28%) of the total energy dissipated during each turn [11]. Clearly, it is important to distinguish between energy losses due to ski–snow friction and those due to aerodynamic drag, especially in connection with the technical disciplines slalom and giant slalom.

The equation for aerodynamic drag ($F_d$) developed by Lord Rayleigh—$F_d = C_d \cdot \rho \cdot S \cdot V^2 / 2$—incorporates four parameters: the drag coefficient ($C_d$), the skier's frontal cross-sectional area (S), the air density ($\rho$), and wind (skiing) speed (V) (Table 1). Although V is a key determinant of the magnitude of $F_d$, the drag can be reduced significantly by reducing $C_d$ and S through adopting an optimal posture. In one of the first studies on $F_d$ during alpine skiing, two subjects exhibited $C_d$ values of 1.0–1.2 and 1.3–1.4 in an upright and semi-squatting position, respectively, and S values of 0.8–1.3 and 0.6–0.8 $m^2$ in an upright and semi-squatting position, respectively [5]. The values of these same parameters for downhill alpine skiers were considerably lower [12,13], particularly for speed skiers, for whom $C_d$ could be as low as 0.16 [9].

**Table 1.** The variables examined and their abbreviations.

| Variable/Abbreviation | Description |
| --- | --- |
| $C_d$ | Coefficient of aerodynamic drag |
| $C_d \cdot S$ | Product of the coefficient of aerodynamic drag and cross-sectional area |
| CV | Coefficient of variation |
| $F_d$ | Aerodynamic drag |
| H | Body height |
| h | Shoulder height, i.e., the distance between the shoulders and ground. |
| $h_n$ | Relative shoulder height = h/H |
| LoA | Limits of agreement |
| $S_r$ | Reference cross-sectional area |
| S | Skier's cross-sectional area |
| $S_n$ | Normalized cross-sectional area = $S/S_r$ |
| V | Relative wind speed |
| $\rho$ | Air density |

The studies performed in wind tunnels focused primarily on optimizing posture to reduce aerodynamic drag, examining not only $C_d$ and S, but also the product $C_d \cdot S$ as a function of posture [10,12,14,15]. Their findings clearly demonstrate that posture height and joint angles, as well as the positioning of the arms and head, all influence aerodynamic drag. Although computer simulation offers a potential alternative to measurements in wind tunnels, to our knowledge, in the case of alpine skiing, this approach has only been used to investigate the effects of wind on various body postures during the flight phase of downhill skiing jumps [16].

Recently, Elfmark and colleagues [14] concluded that virtually any reduction in aerodynamic drag improves skiing performance significantly. Technological advances, and in particular Global Navigation Satellite Systems (GNSS), now allow parameters related to performance to be monitored along an entire ski course [17]. In an attempt to achieve a more detailed analysis of performance that also includes energy losses due to aerodynamic drag and ski–snow friction, an individualized experimental model has been combined with GNSS measurements to estimate the aerodynamic drag on a skier in different positions [11]. Although this approach is potentially of considerable value, it requires measurements in different skiing positions and at different wind speeds in expensive wind tunnels prior to the measurements on-snow, which limits its applicability in practice. Such measurements in a wind tunnel could be avoided if a reliable model of the aerodynamic drag on skiers performing the various technical disciplines in different positions were available.

To our knowledge, because of the nature of the postures investigated, models of this kind that are presently available are applicable to the speed disciplines only [12,14,15] or

are based on monitoring the trajectories of multiple body segments of recreational skiers, which requires more complex technology [10]. Accordingly, the present investigation was designed to develop and assess a model of the aerodynamic drag on elite alpine skiers competing in the technical disciplines in order to enhance analyses of performance based on GNSS data alone.

## 2. Materials and Methods

### 2.1. Participants

Ten male members of the Swedish alpine ski team (age: 19.9 ± 3.6 yrs; height: 1.81 ± 0.05 m; body mass: 80.8 ± 7.0 kg; International Ski Federation points in giant slalom: 27.3 ± 13.1 (means ± SD)) volunteered and signed their written consent to participate. This study was approved by the Regional Ethics Committee in Trondheim, Norway.

### 2.2. Experimental Setup and Measurement Procedures

The experiments were carried out in a wind tunnel (12.5 m long, 2.7 m wide, and 1.8 m high) at the Norwegian University of Science and Technology (NTNU) in Trondheim, Norway, which was equipped with a 220 kW centrifugal blower capable of generating wind speeds of up to 25 m/s. Aerodynamic drag was measured using a force platform (model 9286AA, Kistler Instrument Corp., Winterthur, Switzerland). Video films recorded in the sagittal plane with a Sony HDR-HC7 high-resolution camcorder (Sony Corp., Tokyo, Japan) allowed determination of the skier's height at the shoulder relative to the ground utilizing two-dimensional kinematics software (Avi AD Measure v2.4, Intelligent Solutions and Consulting s.p., Kranjska Gora, Slovenia). In addition, another Sony HDR-HC7 camcorder filmed the skier from behind (Figure 1) and click-based interactive segmentation [18] of the images extracted from this film was employed to obtain the skier's "frontal" cross-sectional area (S). To obtain his reference cross-sectional area ($S_r$), each skier was subjected to an iDXA scan (Lunar iDXA, GE Healthcare, Madison, WI) prior to the measurements in the wind tunnel and these images were processed using the same interactive segmentation.

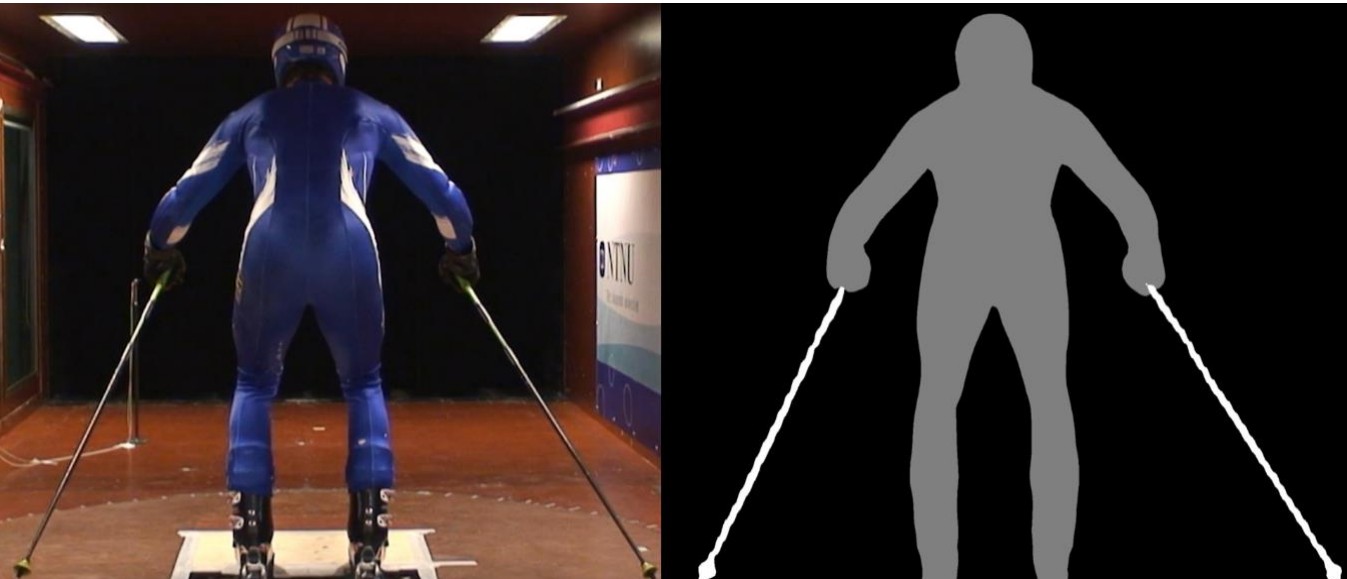

**Figure 1.** A skier in the wind tunnel filmed from behind (**left**) and the segmentation mask (**right**) used to determine his cross-sectional area.

During the measurements in the wind tunnel, each skier wore the entire gear—racing suit, ski boots, gloves, helmet, goggles, and poles— that he wears when competing. Prior to these measurements, the difference between the kinesthetic sensation of anterior-posterior balance in the wind tunnel and on-snow was explained and demonstrated to the skiers.

Thereafter, each participant was required to adapt his skiing posture to the conditions in the wind tunnel.

A force plate, to which ski bindings were attached and adjusted individually with respect to the distance between the legs, was used to make the measurements. In random order, the skier adopted (in response to a visual signal) one of three different standard positions (high, middle, tuck), with exposure five times in each position to winds of 40, 60, and 80 km/h. These positions correspond to those assumed during giant slalom competitions, as determined by recording two subjects during such a competition. The high position is adopted during weight transfer, the medium during turning, and the tuck position at the highest speeds with less extensive turning.

To compensate for the potential recording of forces due to the skier's body weight, the force plate signal was adjusted to zero in the absence of any wind prior to measurement at each different speed and for each individual skier. The aerodynamic drag ($F_d$) was determined from the component of force in alignment with the direction of the wind, utilizing the average of 5 s of continuous values recorded after a stable signal was achieved.

*2.3. Calculations*

The product of the skier's cross-sectional area (S) and the coefficient of aerodynamic drag ($C_d$) was derived from the Rayleigh drag equation $C_d \cdot S = 2 \cdot F_d / (\rho \cdot V^2)$, where $F_d$ is aerodynamic drag, $\rho$ is the air density, and V is the wind speed. Thereafter, the cross-sectional area (S) derived from image processing (see above) allowed the direct calculation of $C_d$. To create experimental models that can be used more generally, dimensionless variables were introduced. For this purpose, the skier's cross-sectional area ($S_n$) was defined as $S_n = S/S_r$ and, analogously, his normalized shoulder height $h_n = h/H$, where h is the distance of his shoulder above the ground and H is his body height.

*2.4. Statistical Analyses*

All data were assessed for normality employing the Kolmogorov–Smirnov test. In cases where the data were not distributed normally, the non-parametric Bland–Altman plot was applied for statistical analysis. To test for differences between parameters, one-way ANOVA with repeated measures was utilized. The application of Mauchly's W test indicated that the assumption of sphericity was violated in two cases, and this was corrected on the basis of the epsilon value obtained with either the Huynh–Feldt or Greenhouse–Geisser procedure. For post hoc analysis, paired-sample *t*-tests were employed to evaluate differences between parameters at different wind speeds. In the case of the regression models, the most common linear regression (least squares-fit) was created utilizing the Curve Fitting Toolbox (MatLab, Mathworks, Natick, MA, USA). All values are expressed as means $\pm$ standard deviations and a *p*-value $< 0.05$ was considered statistically significant.

**3. Results**

The first step in the creation of our model involved the determination of each skier's coefficient of aerodynamic drag ($C_d$), the product of this coefficient and his cross-sectional area ($C_d \cdot S$), and his normalized shoulder height ($h_n$) and cross-sectional area ($S_n$) at different wind speeds in the three representative skiing positions (Table 2). $C_d$ in the high position differed significantly at wind speeds of 40 and 80 km/h, with only a tendency towards a difference between 60 and 80 km/h (Table 2). In addition, with the high and middle positions, there was a trend toward different $h_n$ values at 40 vs. 80 km/h, as well as at 60 vs. 80 km/h, whereas $S_n$ and the product $C_d \cdot S$ were independent of wind speed.

The experimental data from the wind tunnel could be used to develop two experimental models as a function of normalized shoulder height $h_n$ that could be employed in combination with the GNSS data to solve the Rayleigh drag equation. The $C_d \cdot S$ for the first of these models and $C_d$ and $S_n$ for the second are presented in Figures 2 and 3, respectively. In both cases, the dependencies of these parameters on $h_n$ in the higher (high and middle) and tuck positions are shown separately. Since the $C_d \cdot S$, $C_d$, and $S_n$ data, as functions of

$h_n$, exhibited a breakpoint, two separate linear regressions for each of these variables were required for a better fit of the data. The linear regression functions obtained with the first experimental model were $Cd \cdot S = 1.795 \cdot h_n - 0.760$ for the high and middle positions and $0.584 \cdot h_n - 0.045$ for the tuck position (Figure 1). In the case of the second model, $Cd = 1.930 \cdot h_n - 0.427$, $S_n = 1.141 \cdot h_n + 0.094$ for the high and middle positions and $Cd = 1.071 \cdot h_n + 0.241$, $S_n = 0.605 \cdot h_n + 0.222$ for the tuck position (Figure 2).

**Table 2.** Comparison of the coefficient of aerodynamic drag ($C_d$), the product of this coefficient, and the skier's cross-sectional area ($C_d \cdot S$) and normalized shoulder height ($h_n$) and cross-sectional area ($S_n$) at three different wind speeds (V = 40, 60, and 80 km/h) in the three skiing positions (high, mid, and tuck).

| Position | Parameter | V = 40 km/h | V = 60 km/h | V = 80 km/h | *p*-Sphericity | ANOVA F Value | Paired Sample *t*-Tests |
|---|---|---|---|---|---|---|---|
| High | $C_d$ | 1.17 ± 0.09 | 1.15 ± 0.09 | 1.09 ± 0.08 | 0.105 | 8.24 | 40 vs. 80, 60 vs. 80 [‡] |
| | $C_d \cdot S$ [m$^2$] | 0.66 ± 0.09 | 0.66 ± 0.09 | 0.63 ± 0.09 | 0.121 | 3.30 | NA |
| | $h_n$ | 0.78 ± 0.02 | 0.79 ± 0.02 | 0.78 ± 0.02 | 0.041 * | 4.57 | 40 vs. 60 [‡], 60 vs. 80 [‡] |
| | $S_n$ | 0.97 ± 0.05 | 0.99 ± 0.05 | 0.98 ± 0.05 | 0.032 * | 2.19 | NA |
| Middle | $C_d$ | 1.00 ± 0.09 | 0.98 ± 0.09 | 0.94 ± 0.07 | 0.791 | 6.99 | 40 vs. 80 [‡], 60 vs. 80 [‡] |
| | $C_d \cdot S$ [m$^2$] | 0.55 ± 0.07 | 0.53 ± 0.08 | 0.51 ± 0.07 | 0.592 | 2.24 | NA |
| | $h_n$ | 0.70 ± 0.03 | 0.72 ± 0.03 | 0.70 ± 0.03 | 0.399 | 3.96 | 40 vs. 60 [‡], 60 vs. 80 [‡] |
| | $S_n$ | 0.90 ± 0.07 | 0.92 ± 0.07 | 0.92 ± 0.06 | 0.840 | 2.47 | NA |
| Tuck | $C_d$ | 0.80 ± 0.04 | 0.79 ± 0.05 | 0.75 ± 0.05 | 0.132 | 5.45 | 40 vs. 80 [‡], 60 vs. 80 [‡] |
| | $C_d \cdot S$ [m$^2$] | 0.24 ± 0.02 | 0.24 ± 0.03 | 0.23 ± 0.03 | 0.696 | 2.66 | NA |
| | $h_n$ | 0.48 ± 0.03 | 0.48 ± 0.03 | 0.48 ± 0.03 | 0.483 | 1.19 | NA |
| | $S_n$ | 0.51 ± 0.07 | 0.51 ± 0.03 | 0.52 ± 0.03 | 0.446 | 1.44 | NA |

The values presented for each parameter are means ± standard deviations. * Sphericity adjusted (GG: Eps < 0.75 HF: Eps > 0.75), [‡] trend ($0.05 \leq p \leq 0.1$); NA, not applicable, i.e., the ANOVA resulted in a *p*-value > 0.05.

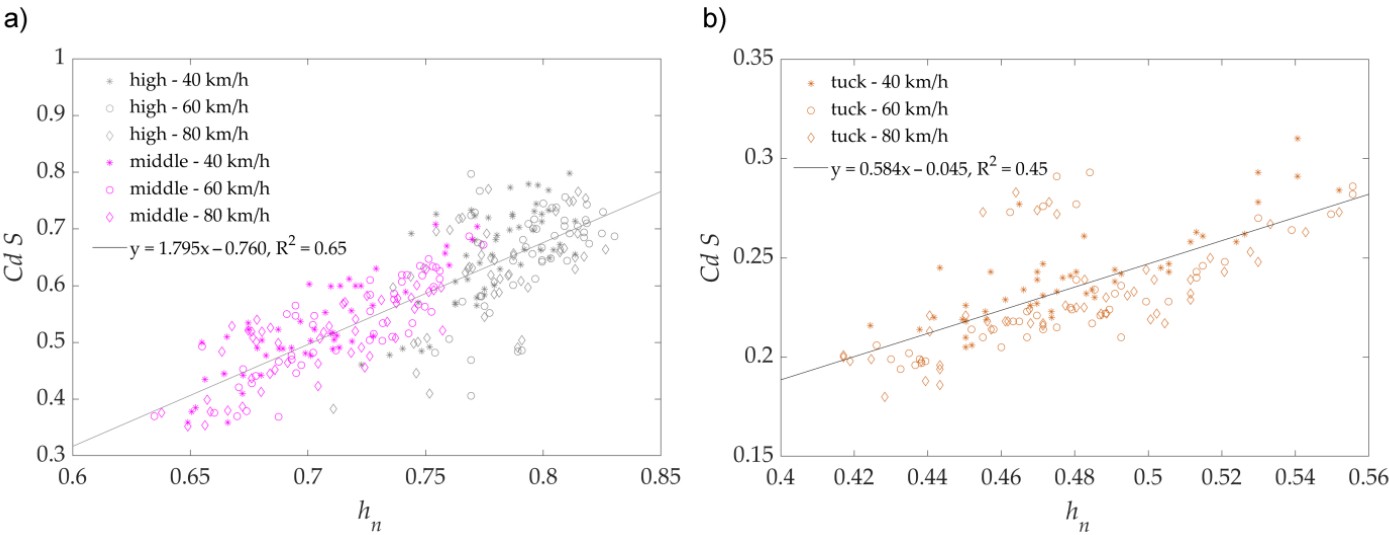

**Figure 2.** Linear regression model with R$^2$ values for the product of the coefficient of aerodynamic drag and cross-sectional area ($C_d \cdot S$) as a function of normalized shoulder height ($h_n$) (**a**) in the high (violet) and middle positions (grey)); (**b**) in the downhill (tuck) position (brown).

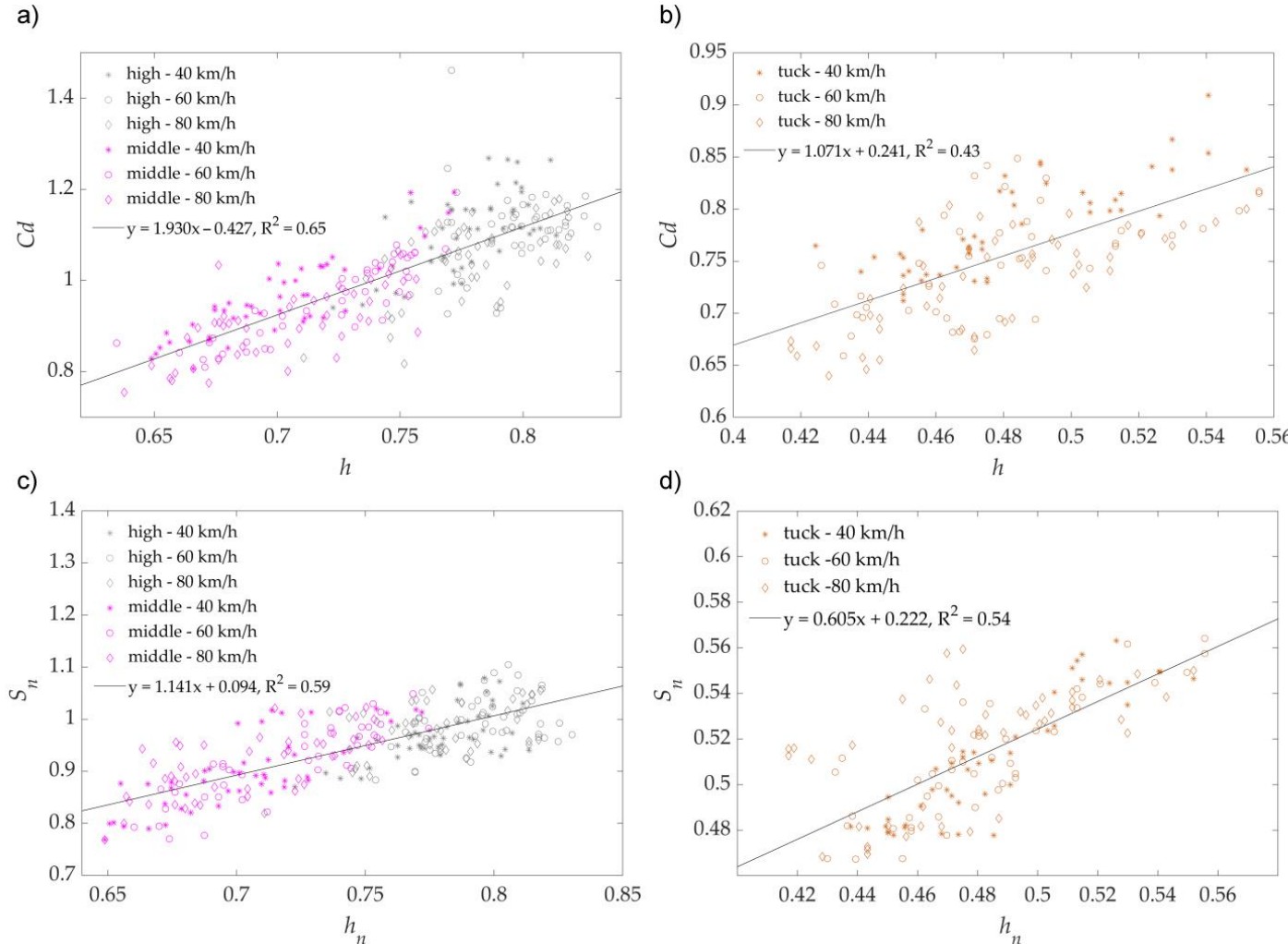

**Figure 3.** Linear regression model with $R^2$ values for the coefficient of the aerodynamic drag ($C_d$) and normalized cross-sectional area ($S_n$) as a function of normalized shoulder height ($h_n$): (**a**,**c**) in the high (violet) and middle positions (grey)), (**b**,**d**) in the downhill (tuck) position (brown).

Figure 4 depicts the deviations between the actual measurements of parameters and the values calculated utilizing these regression models in the form of Bland–Altman plots along with the corresponding coefficients of variation (CV). The magnitude of the 95% limits of agreement (LoA) for all three parameters ($C_d \cdot S$, $C_d$ and $h_n$) ranged between 0.02 ($S_n$) and 0.15 ($C_d$), while the CV ranged from 3.9% ($S_n$) to 7.7% ($C_d \cdot S$).

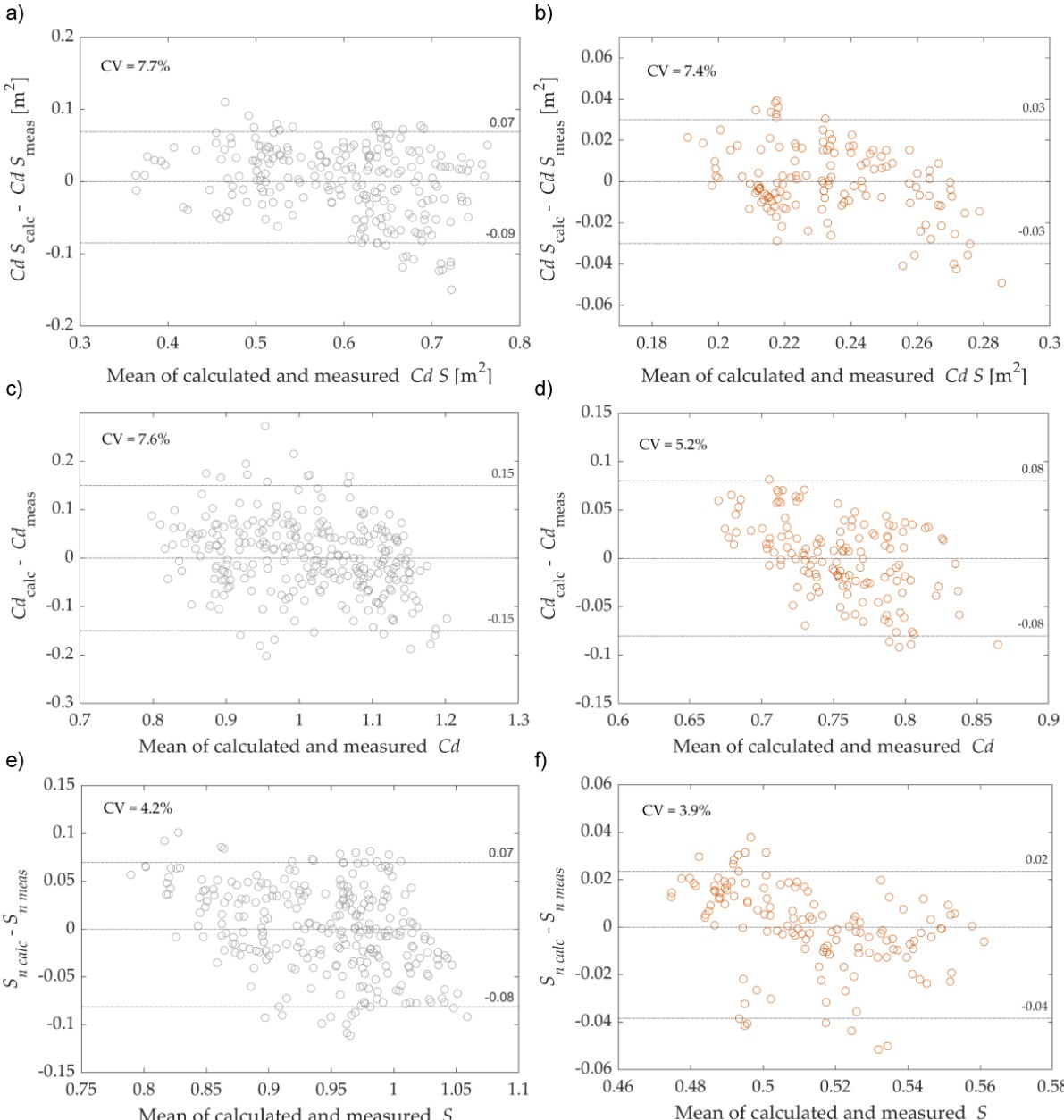

**Figure 4.** Bland–Altman plots demonstrating the mean differences between the calculated and measured values for the three parameters of interest. (**a**,**b**): product of the coefficient of aerodynamic drag and cross-sectional area ($C_d \cdot S$), (**c**,**d**): coefficient of aerodynamic drag ($C_d$), and (**e**,**f**): the normalized cross-sectional area ($S_n$). The diagrams on the left (**a**,**c**,**e**) depict these differences in the high and middle positions, while those on the right (**b**,**d**,**f**) are for the downhill (tuck) position. The dashed lines represent the limits of agreement. CV = coefficient of variation.

## 4. Discussion

The major finding here was that the two experimental models we developed provide values that enable valid monitoring of the aerodynamic drag on elite alpine skiers performing the technical disciplines. Both models involve functions of the normalized shoulder height $h_n$, with the first model estimating the product of the coefficient of aerodynamic drag and cross-sectional area ($C_d \cdot S$) and the second estimating the coefficient of aerodynamic drag $C_d$ and the normalized cross-sectional area of the skier $S_n$. In addition, normative values for $C_d$, $S_n$, $h_n$, and $C_d \cdot S$ in the three positions (high, middle, and tuck), typically

adopted by elite skiers when performing the technical disciplines at three different wind speeds (40, 60, and 80 km/h), were obtained.

These models were based on monitoring 10 elite skiers in three different standard positions in a wind tunnel, assuming that each skier adopted these positions appropriately. However, even if some skiers, despite their high level of skill, took somewhat different positions, this variation actually provided more continuous data for calculating our models. Although it would have been possible to require our skiers to take strictly defined optimal positions, as in certain other studies [12,14,15], in reality, the positions of elite skiers in the technical disciplines often deviate from optimal as they adapt their turns to a variety of gate setups, as well as varying terrain and snow conditions [17].

### 4.1. Normative Values

We found that the average coefficient of aerodynamic drag $C_d$ in the high, medium, and tuck positions was 1.09–1.17, 0.94–1.00, and 0.75–0.8, respectively (Table 1). Similar to a previous report [12], we observed significant differences in these values for $C_d$ in the different positions. However, the $C_d$ values determined here were considerably lower than those reported for downhill skiers (mean reference value: 0.725) or speed skiers (range: 0.15–0.167) [9]. The corresponding average normalized cross-sectional areas ranged from 0.97–0.99, 0.90–0.92, and 0.51–0.52. With the exception of $C_d$ in the high position at wind speeds of 40 vs. 80 km/h, our variables demonstrated some tendency toward, but no statistically significant dependence on, wind speed. These observations indicate that, even though the same position at all three wind speeds was instructed, the positions assumed by some of the skiers may have differed slightly.

On average, $C_d \cdot S$ ranged from 0.63 to 0.66, 0.51 to 0.55, and 0.23 to 0.24 in the high, middle, and tuck positions, respectively. Despite the significant difference in or observed trends towards differences in $C_d$ (Table 2), the product $C_d \cdot S$ was independent of wind speed. Interestingly, the $C_d \cdot S$ values obtained here were considerably higher than those reported for speed disciplines (range: 0.15–0.4) [14,15], clearly confirming the necessity of monitoring and/or modelling aerodynamic variables for the technical and speed disciplines separately.

### 4.2. Aerodynamic Drag Models

Since the bodies of individual skiers differ, we chose to express data as dimensionless, relative/normalized values wherever possible, in order to allow a more general application of our experimental models (Figures 2 and 3). In contrast to previous findings in connection with downhill skiing [12,14,15], our present data indicate clearly that when applying models of aerodynamic drag to the technical disciplines, it is necessary to distinguish between the positions taken during more intense turns (which are more frequent in the technical disciplines) and those adopted during less intense turns or on flat terrain. In these disciplines, transitions from one position to another usually occur quite rapidly; during the transition from a higher to the tuck position and vice-versa, $h_n$ can be estimated here to be $\cong 0.6$, which is almost exactly the mid-point of the individual values for the higher and tuck positions (Figures 2 and 3).

Our first model allows direct calculation of the product $C_d \cdot S$ from the relative shoulder height $h_n$ using two linear regression functions (Figure 2). If the skier is equipped with a global navigation satellite system (GNSS) with the antenna positioned at shoulder height [17], this model, in combination with skiing speed, provides the aerodynamic drag directly from the Rayleigh equation. Thus, in this case, only the relative shoulder height (corresponding to the absolute distance between the shoulders and skis) and skiing speed need to be measured or calculated. A calculation of the product $C_d \cdot S$ as a function of neck height or various joint angles was common in previous investigations on downhill skiing [12,14,15] and, in particular, the relationship to joint angles provides deeper insight into the effects of posture on aerodynamic drag. However, the determination of joint angles as independent variables requires equipment that is more technologically complex.

Since this simpler experimental model calculates the absolute value of the product $C_d \cdot S$ from the relative height $h_n$ alone, this approach might be expected to provide less reliable values for skiers whose height differs from those of our subjects. Therefore, we created another experimental model that, in addition to $C_d$, incorporates the relative cross-section $S_n$, which is also dependent on $h_n$. On average, this latter model resulted in lower limits of agreement (LoA) and coefficient of variation (CV) (Figures 2 and 3). Here, the reference cross-sectional area ($S_r$) of each skier was obtained from iDEXA images, which are often available for elite skiers. Alternatively, when such scans are not available, the skiers can be photographed in an upright and extended position against a wall using a conventional camera and $S_r$, subsequently obtained by processing the resulting image with the same click-based interactive segmentation utilized here [18].

In general, the determination of the aerodynamic drag associated with skiing utilizing our models involves an error of 3.9–7.7% (CV). On the other hand, the division of the LoA (Figures 2 and 3) for each parameter by the average value for this same parameter to obtain a relative error that includes 95% of the values measured results in a range of 4.5–16.5%. These levels of accuracy are only slightly poorer than those reported previously utilizing a model requiring more complex three-dimensional measurements of multiple segments of the skier's body [10].

By employing our new experimental models in combination with the methodology described in a previous report [11], the energy dissipation caused by aerodynamic drag can be determined as a numerical integration of the product of $F_d$ and the distance travelled. Subsequently, these energy losses can be divided by the total energy losses to obtain the relative energy loss associated with aerodynamic drag. The highest deviation with our model is expected to be 16.5%, and, on average, only 15% of energy losses associated with giant slalom (and even less in the case of slalom) are due to aerodynamic drag [11]. Therefore, the uncertainty in the calculation of energy losses associated with aerodynamic drag relative to total energy with our models is expected to be less than 2.5%. This low level of uncertainty allows reliable determination of the fraction of energy loss due to friction when performing the technical disciplines, which helps athletes to know whether their energy losses are due primarily to their aerodynamic position or guidance of the skis (technique). Such information can provide novel insights into the effects of different gate settings and terrain and snow conditions.

*4.3. Limitations*

This current investigation is associated with limitations similar to those encountered in previous studies of a similar nature [10,14,15]. Although our elite skiers possess excellent body awareness and are used to automatically assuming certain positions, it is possible that, in the wind tunnel, they did not adopt exactly the same positions as they normally do when racing. In an attempt to minimize this limitation, the head coach of the national team examined each skier's positions prior to measurement in the wind tunnel and informed him whether any adjustment was necessary.

An additional limitation was that our models did not consider all the possible positions that can be adopted, particularly those that may deviate significantly from the standard positions due to, e.g., particular snow/course conditions and/or loss of balance. At the same time, since such deviant positions tend to be relatively short-lived, their contribution to the entire turn should be quite small. Furthermore, the positions adopted in the wind tunnel were symmetrical and, in practice, skiers adopt asymmetrical positions, which are difficult to simulate appropriately in a wind tunnel. Moreover, movements during skiing are dynamic, while for practical reasons, the positions monitored in the wind tunnel were static. Finally, the models we developed here are based on data obtained with male skiers and it remains to be seen how valid they are for female skiers.

## 5. Conclusions

Here, in order to help alpine skiers improve their performance, we developed experimental models designed to determine the aerodynamic drag associated with skiing the technical disciplines. For simpler analyses, normative values for the three different positions can be employed, while when greater accuracy is desired, two different models that both distinguish between the higher and tuck positions ($[C_d \cdot S/h_n]$ and $[C_d / h_n] \cdot [S_n/h_n]$) with an accuracy of 4.5–16.5% can be applied. With these latter models, the uncertainty in the determination of energy losses associated with aerodynamic drag relative to the total loss of energy is expected to be <2.5%. In combination with GNSS measurements, this provides valuable information concerning potential differences in such mechanical energy losses associated with friction and aerodynamic drag when performing slalom and giant slalom.

**Author Contributions:** Conceptualization, M.S. and H.-C.H.; methodology, M.S., H.-C.H. and N.V.; formal analysis, N.V., J.O. and M.S.; data collection and handling, M.S., H.-C.H., J.O. and M.M.; writing—preparation of the first draft, M.M. and M.S.; writing—review and editing, all authors; visualization, N.V. and M.S.; supervision, M.S.; project administration, M.S.; funding acquisition, M.S. All authors have read and agreed to the published version of the manuscript.

**Funding:** This research was funded by the Slovenian Research Agency (P5-0147).

**Institutional Review Board Statement:** "The study was conducted in accordance with the Declaration of Helsinki, and approved by the Regional Ethics Committee in Trondheim, Norway, 2009".

**Informed Consent Statement:** Informed consent was obtained from all subjects involved in the study.

**Data Availability Statement:** The data presented in this study are available on request from the corresponding author providing the access does not interfere with the conditions provided by the ethics committee.

**Acknowledgments:** The authors would like to sincerely thank all the skiers and their coaches for their helpful cooperation. Many thanks also to Lars Saetran and Luca Oggiano for their help with the collection of data.

**Conflicts of Interest:** The authors have no conflict of interest to declare.

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
