# Peer review of "Application of Experimental Measurements in a Wind Tunnel to the Development of a Model for Aerodynamic Drag on Elite Slalom and Giant Slalom Alpine Skiers"

_applsci, doi:10.3390/app12020902_

Round 1

Reviewer 1 Report

see attached file

Author Response

Reviewer 1

The article presents data and regression models for the drag area and the drag coefficient for varying position and speed of elite alpine skiers. Thus, the data are valuable. However the manuscript needs serious overworking.

Response to reviewer: Taking your comments and suggestions, as well as those of the other two reviewers, into consideration, we have now revised our manuscript extensively.

Main issues:

The work provides wind tunnel data of alpine skiers and derives regression equations for CdA and Cd. Additionally, the authors give some statistical evaluation of their results. Nothing else was presented in the methods and results sections. The whole manuscript should be overworked with respect to this fact. The authors write about many things, which are not related to the presented results.

Response to reviewer: In addition to expressing Cd, Sn and CdS as a function of hn, this work provides reference values for the three different skiing positions at different wind speeds, as well as the deviations (in the form of Bland-Altman plots and CVs) between actual measurements of parameters and the values calculated utilizing these experimental models.

We have now gone through the manuscript and omitted information in the Introduction and Discussion concerning the use of IMU, force plates, and, to a certain extent, GNSS, that is of lesser relevance to our Results. Moreover, we have added information related to previous investigations on aerodynamic drag.

Abstract: The basic result of the paper are results on CdA and Cd. These are missing in the abstract.

Response to reviewer: We now include concrete values for Cd, CdS, Sn and Hn in the Abstract.

Introduction: The introduction should motivate why the selected research question was considered and lead to the purpose of the work. To my opinion, the purpose of the work was to measure and calculate regression equations of drag area and drag coefficient.

Response to reviewer: Indeed, our main purpose was to develop and assess regression models for determination of CdS, Cd and Sn designed to enhance analysis of performance in the technical disciplines of alpine skiing. This is now explained more extensively in the Introduction to motivate our selection of research question.

The authors can write about the importance of drag, drag area, and drag coefficient for the performance in the various skiing disciplines. However, I do not understand how GNSS measurements are related to this. Maybe, the authors can write in the discussion how they want to apply the presented drag measurements in GNSS measurements.

Response to reviewer: GNSS technology has been utilized previously in combination with measurements in a wind tunnel to calculate the aerodynamic drag associated with alpine skiing (Supej et al. 2013). However, such measurements in wind tunnels are expensive and, therefore, not readily accessible in many cases. Accordingly, we aimed to develop and evaluate regression models for the aerodynamic drag associated with the technical disciplines of alpine skiing based on GNSS data alone, i.e., without having to make measurements in a wind tunnel.

As now explained in the text, other models for evaluating aerodynamic drag have also been developed (Meyer et al., 2012; Elfmark et al., 2018, 2021; Barelle et al., 2004; Thompson et al., 2001), but these either require measurements that are technologically more complex and/or focus only on the tucked or semi-tucked positions utilized to the greatest extent in downhill or speed skiing.

Missing is the discussion (benefits, faults, dependencies…) of previous attempts to measure and model drag – these works are just briefly mentioned. The purpose has to be reformulated.

Response to reviewer: In addition to clarifying our purpose in greater detail, we have now added information concerning such previous attempts to both the Introduction and Discussion.

Methods: I would add a brief paragraph on the calculation of the regression equations. Maybe another paragraph on statistics.

Response to reviewer: Additional information concerning how the regression equations were obtained is now included in the section on statistical analysis in the Methods.

Results: Since the regression equations are a main result they should appear in the text. Additionally, the authors should explain why the give a regression equation for high and middle positions and another for low positions.

Response to reviewer: We have included the regression equations in the Results and explain why there are two distinct regression models for the high and middle positions and another for the low (tuck) position.

Discussion: The discussion is too long and should use a more precise and compact formulation. The authors discuss their own findings about drag (CdA, Cd, relation to Sn and hn) but do not relate them to findings of other authors. They also do not discuss differences to other authors.

Response to reviewer: We have now revised the Discussion to be both more concise and precise, as well as to compare our findings in greater detail to those of other studies. For example, we have deleted less relevant information concerning inverted pendulum modelling and machine learning. Finally, subheadings have been added to make the organization of the Discussion even clearer.

Conclusion: A conclusion should be short and direct. Delete the second paragraph.

Response to reviewer: The conclusion is now shorter (i.e., the second paragraph has been deleted, as suggested) and more direct.

Writing: The writing should be improved. Better English. More precise. More compact.

Response to reviewer: The writing in the entire manuscript has now been re-edited by a native English speaker.

Citations: Massive self-citation, 7 of 17 citations are of Supej et al. Several citations are not related to this work others should be added.

Response to reviewer: Three of the citations related to “Supej” have now been deleted and four new citations of direct relevance to aerodynamic drag added.

Specific remarks: Select a title, which describes what you are doing in the work!

Response to reviewer: The title has been revised to reflect more effectively what this work is about.

Affiliations: delete 5), I guess Holmberg works at 3 or 4, not both and Supej in 1 or 2, not both.

Response to reviewer: Actually, these two authors are each affiliated with two different institutions. Affiliation 5 has now been deleted.

Abstract: wordings “experimental models” better regression model

Response to reviewer: We have replaced “experimental” with “regression” in the Abstract.

Intro: “inverted pendulum modelled” use “inverted pendulum” or “inverted pendulum model”

Response to reviewer: This section has been removed.

Intro: “[13,14] (Berel, Ruby … [15])” … reformulate

Response to reviewer: Reformulated.

Intro: Reformulate last paragraph! State purpose!

Response to reviewer: This section has been revised to state our purpose more precisely.

Sec 2.2: name the wind tunnel (name, location, state)

Response to reviewer: Information included as suggested.

Sec 2.3: Sr is not defined!

Response to reviewer: Actually, Sr is defined in section 2.2. In response to a comment by reviewer 2, we also include a table that lists the variables of interest and their abbreviations.

Results: “In addition there was a trend towards different hn …”: I do not understand this sentence. Hn is selected by the skier.

Response to reviewer: This sentence has now been revised.

Results, last paragraph p4: Some parts of this paragraph should be moved to the methods section.

Response to reviewer: This section has been revised to focus solely on the Results.

Discussion: On p7 CV and LoA are not defined.

Response to reviewer: In response to this comment, we have added these definitions. As mentioned above, we also include a table listing the variables examined and their abbreviations.

Discussion: “2.5%” this needs citation

Response to reviewer: The value of 2.5% comes from the reference cited earlier in this same sentence, which we now make clearer.

Reviewer 2 Report

The manuscript entitled “Experimental Models for Reliable Determination of the Aero-dynamic Drag Associated with Performance of the Technical Disciplines by Elite Alpine Skiers” by Majerič et al. has been reviewed. It is a nice piece of work that will be of interest to the alpine skier’s community. The authors did perform experimental analysis to monitor and estimate the aerodynamic drag on a skier in different positions.

A few minor suggestions could help this work become even better. Therefore, I only recommend some minor revisions. I recommend this paper be accepted after the following minor concerns are addressed.

The introduction needs to be revised to reflect the major findings from previous investigations, and more details about the results of the previous findings need to be mentioned.

There are many grammatical errors, improper wording, and the figures are not well prepared. Therefore, all figures need to replace with high resolution and quality ones.

The determination of the fraction of energy losses, including uncertainty, from experimental measurements, needs further explanation/clarification.

There are many acronyms, abbreviations, and symbols in the manuscript. Therefore, a nomenclature would be helpful.

Author Response

Reviewer 2

The manuscript entitled “Experimental Models for Reliable Determination of the Aero-dynamic Drag Associated with Performance of the Technical Disciplines by Elite Alpine Skiers” by Majerič et al. has been reviewed. It is a nice piece of work that will be of interest to the alpine skier’s community. The authors did perform experimental analysis to monitor and estimate the aerodynamic drag on a skier in different positions.

Response to reviewer: Thank you, we appreciate these positive comments.

A few minor suggestions could help this work become even better. Therefore, I only recommend some minor revisions. I recommend this paper be accepted after the following minor concerns are addressed.

Response to reviewer: We have made changes in response to your suggestions and concerns, as well as those of the other two reviewers.

The introduction needs to be revised to reflect the major findings from previous investigations, and more details about the results of the previous findings need to be mentioned.

Response to reviewer: The Introduction has been revised as suggested to describe in greater detail the major findings of previous investigations.

There are many grammatical errors, improper wording, and the figures are not well prepared. Therefore, all figures need to replace with high resolution and quality ones.

Response to reviewer: The entire manuscript has been re-edited by a native English speaker. The figures are now of higher resolution and in the .tif format.

The determination of the fraction of energy losses, including uncertainty, from experimental measurements, needs further explanation/clarification.

Response to reviewer: We now clarify how the fraction of energy losses due to aerodynamic drag is calculated from the experimental measurements. In addition, the sentence concerning uncertainty has been revised for better understanding.

There are many acronyms, abbreviations, and symbols in the manuscript. Therefore, a nomenclature would be helpful.

Response to reviewer: A list of the variables examined and their abbreviations is now presented as Table 1.

Reviewer 3 Report

The authors have presented an experimental model to determine the coefficient of drag for alpine skiers. They have conducted wind tunnel experiments on professional skiers in the three standard positions, followed with statistical analysis of the results. The experimental model has shown some deviation from the experimental results, however, it can be acceptable as a preliminary indication with a simple model using only the normalized height of the skiers.

The manuscript has shown no significant scientific errors and I believe it has a merit and can be interesting to the readers. It is well organized and the methods are stated clearly. I recommend publishing of the manuscript after minor corrections. Some comments are as follows:

  • Using Rayleigh's equation, since the drag coefficient doesn't change significantly with the wind speed, the drag force must have increased significantly to balance the equation (Recall that Cd is proportional to the drag force, and inversely proportional to the square of the wind speed). It can be misleading to readers who may think that wind speed doesn't matter in the mechanical energy loss. Accordingly, the authors must state clearly that although the coefficient of drag decreases, the drag force increases with higher wind speed, and hence more mechanical loss. However, it is more efficient at higher speeds.
  • Figures 2-4 need to be enhanced for a better observation by the readers. Figures have low resolution and it is hard to tell which is which.
  • Minor English proofreading should be done specially for the discussion and conclusions sections. Some prepositions are missing and the discussion section needs better organization.

Author Response

Reviewer 3

The authors have presented an experimental model to determine the coefficient of drag for alpine skiers. They have conducted wind tunnel experiments on professional skiers in the three standard positions, followed with statistical analysis of the results. The experimental model has shown some deviation from the experimental results, however, it can be acceptable as a preliminary indication with a simple model using only the normalized height of the skiers.

Response to reviewer: Thank you, we appreciate this positive assessment.

The manuscript has shown no significant scientific errors and I believe it has a merit and can be interesting to the readers. It is well organized and the methods are stated clearly. I recommend publishing of the manuscript after minor corrections. Some comments are as follows:

Response to reviewer: Thank you again for these kind remarks. We have made changes in response to your comments and suggestions, as well as to those of the other two reviewers.

Using Rayleigh's equation, since the drag coefficient doesn't change significantly with the wind speed, the drag force must have increased significantly to balance the equation (Recall that Cd is proportional to the drag force, and inversely proportional to the square of the wind speed). It can be misleading to readers who may think that wind speed doesn't matter in the mechanical energy loss. Accordingly, the authors must state clearly that although the coefficient of drag decreases, the drag force increases with higher wind speed, and hence more mechanical loss. However, it is more efficient at higher speeds.

Response to reviewer: Thank you for this helpful comment, with which we agree fully. We have now made this clearer in the introduction with the following text: “The equation for aerodynamic drag (Fd) developed by Lord Rayleigh – Fd = Cd∙ρ∙S·V2/2 –incorporates four parameters: the drag coefficient (Cd), the skier’s frontal cross-sectional area (S), the air density (ρ) and wind (skiing) speed (V) (Table 1). Although V is a key determinant of the magnitude of Fd, the drag can be reduced significantly by reducing Cd and S through adopting an optimal posture.”

Figures 2-4 need to be enhanced for a better observation by the readers. Figures have low resolution and it is hard to tell which is which.

Response to reviewer: The figures are now of higher resolution and presented in the .tif format.

Minor English proofreading should be done specially for the discussion and conclusions sections. Some prepositions are missing and the discussion section needs better organization.

Response to reviewer: The entire manuscript has been re-edited by a native English speaker, with particular attention being given to the Discussion and Conclusion. In addition, we have now made the organization of the Discussion clearer by including subheadings.